# Profiling Microbial Communities in Idiopathic Granulomatous Mastitis

**DOI:** 10.3390/ijms24021042

**Published:** 2023-01-05

**Authors:** Seeu Si Ong, Jia Xu, Choon Kiat Sim, Alexis Jiaying Khng, Peh Joo Ho, Philip Kam Weng Kwan, Aarthi Ravikrishnan, Kiat-Tee Benita Tan, Qing Ting Tan, Ern Yu Tan, Su-Ming Tan, Thomas Choudary Putti, Swee Ho Lim, Ee Ling Serene Tang, Niranjan Nagarajan, Neerja Karnani, Jingmei Li, Mikael Hartman

**Affiliations:** 1Women’s Health and Genetics, Genome Institute of Singapore, Agency for Science, Technology and Research, Singapore 138672, Singapore; 2Department of Surgery, Yong Loo Lin School of Medicine, National University of Singapore, Singapore 119228, Singapore; 3Human Development, Singapore Institute for Clinical Sciences, Agency for Science, Technology and Research, Singapore 117609, Singapore; 4Saw Swee Hock, School of Public Health, National University of Singapore, Singapore 117549, Singapore; 5Department of Medicine, Yong Loo Lin School of Medicine, National University of Singapore, Singapore 119228, Singapore; 6Metagenomic Technologies and Microbial Systems, Genome Institute of Singapore, Agency for Science, Technology and Research, Singapore 138672, Singapore; 7Division of Surgical Oncology, National Cancer Centre Singapore, Singapore 169610, Singapore; 8Department of Breast Surgery, Singapore General Hospital, Singapore 169608, Singapore; 9Department of General Surgery, Sengkang General Hospital, Singapore 544886, Singapore; 10KK Breast Department, KK Women’s and Children’s Hospital, Singapore 229899, Singapore; 11Department of General Surgery, Tan Tock Seng Hospital, Singapore 308433, Singapore; 12Lee Kong Chian School of Medicine, Nanyang Technological University, Singapore 308232, Singapore; 13Institute of Molecular and Cell Biology, Agency for Science, Technology and Research, Singapore 138673, Singapore; 14Division of Breast Surgery, Changi General Hospital, Singapore 529889, Singapore; 15Department of Pathology, National University Health System, Singapore 119228, Singapore; 16Department of Surgery, Woodlands Health Campus, Singapore 768024, Singapore; 17Data Hub Division, Bioinformatics Institute, Agency for Science, Technology and Research, Singapore 138671, Singapore; 18Department of Biochemistry, Yong Loo Lin School of Medicine, National University of Singapore, Singapore 119228, Singapore; 19Department of Surgery, University Surgical Cluster, National University Health System, Singapore 119228, Singapore

**Keywords:** idiopathic granulomatous mastitis, metagenomic sequencing, microbiota, 16S rRNA, *Corynebacterium*, MaAsLin 2

## Abstract

Idiopathic granulomatous mastitis (IGM) is a rare and benign inflammatory breast disease with ambiguous aetiology. Contrastingly, lactational mastitis (LM) is commonly diagnosed in breastfeeding women. To investigate IGM aetiology, we profiled the microbial flora of pus and skin in patients with IGM and LM. A total of 26 patients with IGM and 6 patients with LM were included in the study. The 16S rRNA sequencing libraries were constructed from 16S rRNA gene amplified from total DNA extracted from pus and skin swabs in patients with IGM and LM controls. Constructed libraries were multiplexed and paired-end sequenced on HiSeq4000. Metagenomic analysis was conducted using modified microbiome abundance analysis suite customised R-resource for paired pus and skin samples. Microbiome multivariable association analyses were performed using linear models. A total of 21 IGM and 3 LM paired pus and skin samples underwent metagenomic analysis. Bray−Curtis ecological dissimilarity distance showed dissimilarity across four sample types (IGM pus, IGM skin, LM pus, and LM skin; PERMANOVA, *p* < 0.001). No characteristic dominant genus was observed across the IGM samples. The IGM pus samples were more diverse than corresponding IGM skin samples (Shannon and Simpson index; Wilcoxon paired signed-rank tests, *p* = 0.022 and *p* = 0.07). *Corynebacterium kroppenstedtii*, reportedly associated with IGM in the literature, was higher in IGM pus samples than paired skin samples (Wilcoxon, *p* = 0.022). Three other species and nineteen genera were statistically significant in paired IGM pus–skin comparison after antibiotic treatment adjustment and multiple comparisons correction. Microbial profiles are unique between patients with IGM and LM. Inter-patient variability and polymicrobial IGM pus samples cannot implicate specific genus or species as an infectious cause for IGM.

## 1. Introduction

Idiopathic granulomatous mastitis (IGM) has been consistently described as a rare and chronic inflammatory infliction of the breast [1,2]. The condition, while presenting clinically and radiologically with features that may mimic breast malignancy, is benign [1,2]. Only when histopathological examination of the breast tissue for noncaseating granulomatous inflammation in the lobules, coupled with the in-depth investigation of each case eliminating fungal, tuberculous, and other causes of granulomatous mastitis, would IGM be diagnosed [3,4,5]. This complex investigative work-up means IGM is usually only diagnosed in high-resource settings, and by clinicians who have awareness of the differential. Furthermore, without a structured disease registry for proper documentation of IGM, an accurate estimation of the global disease incidence and prevalence is absent. Limited case series and study populations published in literature is affirmed by the anecdotal experiences of the rare nature of the disease [6].

Hypothesised aetiologies of IGM include a possible microbial cause [7]. *Corynebacterium* was first associated with IGM when Taylor et al. (2003) reported coryneform bacteria observed in histological examination of breast tissue and *Corynebacterium* species isolated in cultures from patient specimens [8]. Particularly, *Corynebacterium kroppenstedtii*, *C. amycolatum*, *C. tuberculostearicum*, and *C. accolens* species of *Corynebacterium* have been isolated from IGM patient samples [8,9,10]. This has led to reasonable speculation that *Corynebacterium* species could be involved in the pathogenesis of IGM [7]. Published studies investigating microbial communities in IGM rely on culture-dependent methods to isolate bacteria species from patient pus samples; *Corynebacterium* species were identified from polymerase chain reaction (PCR) and 16S rRNA sequencing of cultured bacteria colonies [8,11]. Pathogens incompatible with traditional culture methods cannot be detected through such methods.

Another benign breast inflammation that presents similarly in clinical settings is lactational mastitis (LM), a common complication in breastfeeding women [12,13]. Diagnosis usually requires targeted ultrasound and histopathological examination of biopsied breast tissue to confirm the benign inflammatory condition, and rule out malignancy [14]. As a similarly benign and inflammatory breast disease, LM provides an appropriate comparison group to IGM. Antibiotics is a similar first-line treatment for both conditions, as is the common practice to prescribe patients presenting in outpatient settings with symptoms similar to patients with IGM and LM (localised symptoms: erythema, warmth, oedema, and tenderness; systemic symptoms: fever, and malaise) with antibiotics [12,15]. The dominant *Staphylococcus* genus reported in the microbiology for breast milk in women affected with LM can be compared against the *Corynebacterium* genus associated with IGM [16,17,18,19,20]. LM will serve as a suitable condition to make metagenomic comparisons against while exploring microbial profiles from IGM patient samples.

Previously published studies identified pathogens from microbial cultures and/or observations in histology [8,9,10]. Yu et al. (2016) subsequently performed 16S rRNA sequencing from IGM pus samples, in a purely descriptive approach without statistical analyses [11]. Our work expands on Yu et al. (2016), with the aim to utilise 16S rRNA sequencing on IGM pus samples as well, while including statistical analyses for a wholesale inspection of the microbiome of IGM patient pus samples. Additionally, our study also aims to profile the corresponding skin microbiome of patients with IGM. The study objective is to identify microbial agents incompatible with traditional culture methods or cannot be observed on histology, with statistical analyses that evaluate the significance of

Comparisons between IGM and LM pus microbial communities;Comparisons between IGM pus and skin microbial communities in the same patient.

Gaining a better understanding of the microbial flora could potentially identify aetiological pathogens that could be targeted in developing detection assays for potentially simplified disease diagnostics. The involvement of such pathogens in the pathophysiology of IGM could also guide clinical disease management through targeted treatment or for monitoring disease progression and prognosis, with the tendency for recurrence in IGM [21,22,23,24].

## 2. Results

### 2.1. Study Population

Paired pus and skin samples from 21 patients with IGM and 3 patients with LM underwent metagenomic analysis (Appendix A). Patients with IGM (median age = 34 years) were slightly older than patients with LM (median age = 31 years), although this difference was not significant (Table 1). Other demographic variables reported are site of recruitment, year of diagnosis, ethnicity, body mass index, and education level (Table 1). Patient characteristics reported were any smoking history, chronic illness diagnosis, autoimmune conditions, previous infectious disease diagnosis, previous cancer diagnosis, and mastitis treatment history prior to sample collection (Table 1). None of the demographic or patient variables were significantly different between patients with IGM and LM. Patients with LM served as controls for patients with IGM in this study.

### 2.2. Comparing IGM vs. LM Microbial Taxonomic Profiles

Taxonomic profiles of IGM and LM pus and skin samples were predominantly examined at the genus level; this is visualised in Figure 1. *Staphylococcus* genus dominates the three LM pus samples (LM1-3): 90.8%, 26.5%, and 38.5%, respectively. Contrastingly, no dominant genus was observed across the IGM pus samples (Figure 1a). Greater inter-patient variability was also observed in the IGM pus samples, as compared to the LM pus samples (Figure 1a). Multidimensional scaling (MDS) of Bray−Curtis distance between bacteria profiles in IGM and LM pus samples did not show obvious clustering separating the two patient groups; however, their Bray−Curtis distancing were significantly different (PERMANOVA, *p* < 0.001) (Figure 2b). It must be noted that the first two MDS of Bray−Curtis distance between breast pus and skin samples for patients with IGM only explains 16.98% and 12.23% of the variability, respectively, and collectively explains 29.21% of the samples’ variability.

No dominant genus is observed across either the IGM or LM skin samples, collectively or as separate groups (Figure 1b). Again, inter-patient variability is considerable (Figure 1a). All four sample types (IGM pus, IGM skin, LM pus, and LM skin) taken collectively, also did not show obvious clustering in MDS of Bray−Curtis distance between bacteria profiles, but, again, the distancing was significantly different (PERMANOVA, *p* < 0.001) (Figure 2a). The paired LM samples did not appear close to their corresponding pair in the MDS. The first two MDS of Bray−Curtis distance between breast pus and skin samples for patients with IGM and LM only explains 14.97% and 13.11% of the variability, respectively, and collectively explain 28.08% of the samples’ variability.

Both Shannon (information statistic) and Simpson (dominance) indices for measuring alpha-diversity in each sample demonstrated higher diversity in IGM pus samples (Shannon = 2.65 [interquartile range, IQR = 1.91–2.92]) than any other sample type (Appendix A); however, this difference was only borderline statistically significant (Wilcoxon signed-rank test, *p* = 0.083). Comparisons between alpha-diversity of paired pus and skin samples from the same IGM patient found that IGM pus samples were statistically more diverse than the corresponding IGM skin sample (Shannon index Wilcoxon paired signed-rank test, *p* = 0.022; Simpson index Wilcoxon paired signed-rank test, *p* = 0.07; Appendix A). This comparison was not significant for paired samples in patients with LM (Shannon index Wilcoxon paired signed-rank test, *p* = 1.000; Simpson index Wilcoxon paired signed-rank test, *p* = 0.75). Differences in alpha-diversity between IGM pus and LM pus, or IGM skin and LM skin samples, were not statistically significant (Appendix A). Medians and interquartile ranges for IGM and LM samples for both diversity indices are presented in Appendix A.

### 2.3. Corynebacterium

Focus was placed on *Corynebacterium* genera given the literature published demonstrating an association with IGM [7,8,9,10,11]. Inter-patient variability for *Corynebacterium* relative abundance was high amongst patients with IGM, for both pus and skin samples (Figure 3). As this genus is commonly found on the skin, it was unsurprising that IGM skin samples *Corynebacterium* relative abundance was 9.4% [IQR = 5.8–30.2%] (Figure 3). However, *Corynebacterium* relative abundance in IGM pus samples was rather similar, at 9.0% [IQR = 3.8–13.0%] (Figure 3).

No obvious trend was observed in genus-level pairwise comparison for *Corynebacterium* relative abundance between paired IGM pus and skin samples (Figure 3); *Corynebacterium* relative abundance was not statistically significant between the paired samples (Wilcoxon paired signed-rank test, *p* = 0.29; paired LM pus and skin samples, *p* = 0.25 [Figure 3; Appendix A]). However, species-level taxonomic analysis for *Corynebacterium kroppenstedtii* relative abundance was significantly different between paired IGM pus and skin samples (Wilcoxon paired signed-rank test, *p* = 0.022 [Appendix A]).

### 2.4. Associations between Sample Types and Metagenomic Features

Microbiome multivariable association between sample type, covariates and microbial metagenomic features between IGM and LM pus samples using general linear models (MaAsLin 2), with correction for multiple comparisons false positive rate, did not find any genus that were statistically different.

However, pairwise analysis found 24 genera were significantly different between paired IGM pus and skin samples: *Ochrobactrum*, *Delftia*, *Anaerobacillus*, *Gordonia*, *Methylobacterium*, *Fusobacterium*, *Streptococcus*, *Achromobacter*, *Sphingobium*, *Alkanindiges*, *Novosphingobium*, *Capnocytophaga*, *Mycobacterium*, *Burkholderia*, *Peptoniphilus*, *Roseomonas*, *Rothia*, *Finegoldia*, *Agrobacterium*, *Anaerococcus*, *Hydrogenophaga*, *Peptostreptococcus*, *Kocuria*, and *Dermabacter*. A total of 19 genera remained significantly different after adjusting for antibiotic treatment (treated vs. not treated), and duration after antibiotics treatment (samples collected less than 2 weeks after treatment vs. more than 2 weeks after treatment): *Ochrobactrum*, *Delftia*, *Anaerobacillus*, *Gordonia*, *Methylobacterium*, *Fusobacterium*, *Sphingobium*, *Alkanindiges*, *Streptococcus*, *Achromobacter*, *Capnocytophaga*, *Mycobacterium*, *Novosphingobium*, *Peptoniphilus*, *Rothia*, *Finegoldia*, *Burkholderia*, *Roseomonas*, *Anaerococcus*. Beta-estimates, standard deviations, crude and adjusted *p*-values, as well as crude and adjusted q-values (multiple comparisons corrected *p*-value) are reported in Table 2.

Table 2 also shows the beta-estimates, standard deviations, crude and adjusted *p*-values, as well as crude and adjusted q-values for the same pairwise analysis performed at species level taxonomy. Six species with significantly different relative abundance between the paired IGM pus and skin samples were *Acinetobacter schindleri*, *Rothia mucilaginosa*, *Lactobacillus iners*, *Corynebacterium kroppenstedtii*, *Roseomonas mucosa*, and *Kocuria rhizophila*. The association remains significantly different after the adjustments for antibiotic treatment (treated vs. not treated), and duration after antibiotics treatment (samples collected less than 2 weeks after treatment vs. more than 2 weeks after treatment), for 3 species: *Acinetobacter schindleri*, *Rothia mucilaginosa*, and *Lactobacillus iners*; *Corynebacterium kroppenstedtii* was borderline significantly different between the paired IGM samples (*p* < 0.001, q = 0.053 [Table 2]).

## 3. Discussion

In the previously published paper about bacteria communities in IGM pus samples by Yu et al., 2016, clinical metagenomic analysis was performed on breast pus samples from patients with IGM [11]. The paper describes varying domination of *Corynebacterium* genera across their 19 patients with IGM, as compared to little to no *Corynebacterium* genera presence in their LM controls [11]. 

Our study enriches this existing body of work by providing relevant patient comparison via the skin sample metagenomic evaluation from the non-infected site. As such, we were able to perform pairwise comparison on the relative abundance of *Corynebacterium* between patient pus sample and their non-infected skin sample. While the Yu et al., 2016, study amplified the hypervariable region 4 (V4) of 16S rRNA gene, our study also expands upon that with the hypervariable regions 2 to hypervariable region 6 (V2–V6) of the 16S rRNA gene amplified (Appendix A). The genus taxonomy level analysis benefits from higher resolution in identifying bacterial populations specifically from the hypervariable regions 2 and 3 of the 16S rRNA gene [25]. Furthermore, Meisel et al., 2016, have previously demonstrated sequencing the V1-3 hypervariable regions of the 16S rRNA gene for skin microbiome surveys, and by extension pus microbial evaluations as well, provides more precise microbial community characterisation [26]. In contrast, amplifying the V4 region of the 16S rRNA gene is more common for protocols developed for characterising microbiota of other habitats, particularly human gastrointestinal sites [26]. Additionally, the V4 region amplified in the Yu et al., 2016, study, is a much shorter and more conserved sequence, compared to the V2–V6 regions amplified in our study [11,27]. Therefore, our study is likely to find taxonomies not previously found in the Yu et al. 2016 study. Larger fragment amplification has also been found to improve sensitivity and specificity of sequences classified across most taxonomic levels [28].

Other differences between our study and the Yu et al. study include sequencing platforms (Ion PGM Hi-Q 400 Sequencing vs. HiSeq4000) and read lengths (200/400 bp vs. 2 × 150 bp paired-end reads) (Yu et al., 2016 study corresponds for the former metrics, our study corresponds to the latter). Last of the comparisons is the Yu et al., 2016 paper has a purely descriptive approach to evaluating the microbial population of IGM and LM pus samples, whereas our study provides statistical evaluation of the microbial genera and species.

Breast inflammation is often categorised as lactational and non-lactational [29]. LM is commonly caused by skin-colonising bacteria [29]. The most common skin-colonising bacteria implicated in LM is *Staphylococcus aureus*, and increasingly Methicillin-resistant *S. aureus* (MRSA) in particular [29]. *Streptococcus pyogenes*, *Escherichia coli*, *Bacteroides species*, and Coagulase-negative *Staphylococci* have also been identified to cause LM [29]. Prior history of mastitis, nipple cracks and fissures, inadequate milk drainage, maternal stress, lack of sleep, tight-fitting bras, and use of antifungal nipple creams are some of the LM risk factors that might provide conducive environment for unhealthy bacteria growth and colonisation [30,31].

On the other hand, IGM aetiology has always been ambiguous. Possible causes of IGM span autoimmune disease, trauma, lactation, oral contraceptive pill use, and hyperprolactinemia, with a growing literature suggesting *Corynebacterium* infection association with the pathogenesis of IGM [7,32]. In addition to the impossibility of collecting pus samples from healthy women, these biological differences make LM a suitable condition to be studied in comparison with IGM. Our study collected both pus and skin samples to highlight the infiltration of aetiological populations from skin to pus in both sample types collected from LM, that would not be expected in IGM. The dominance of *Staphylococcus* in LM pus but not skin samples, coupled with Bray−Curtis distance differences between paired LM samples implies that LM cannot be entirely attributed to bacterial infiltration from skin colonies. This confirms that other risk factors beyond skin breakages and topical contributions play integral roles in the disease mechanism [30,31].

In patients with IGM, without the skin infiltration aetiological mechanism, we expected to see significantly different genus level taxonomy of microbial populations between pus and skin samples. With the increasing association found between *Corynebacterium* and IGM patient samples, as well as the increasing implication of *Corynebacterium* as a causative pathogen of IGM [8,9,10,11], we focused on studying the presence of *Corynebacterium* in our patient samples. Discerning the role of this genus as infectious, coloniser or contaminant in IGM is challenging [9]. *Corynebacterium* is a common population of Gram-positive bacteria in human skin microbiota [7]. The unique combination of presence of polymorphonuclear leukocytes with Gram-positive rods; and sterile tissue in healthy conditions but testing positive for *Corynebacterium* in IGM, could determine that this genus might be responsible [33]. Alternatively, it can be hypothesised that *Corynebacterium* might contribute to a unique skin flora that makes patients more susceptible to IGM, given the absence of significant difference between *Corynebacterium* relative abundance in paired pus and skin samples. Contrastingly, there could be unique species of the *Corynebacterium* genus that contribute different roles to the disease: the significantly higher *Corynebacterium kroppenstedtii* abundance in pus samples compared to their corresponding paired skin sample (Table 2; Appendix A) could indicate this individual species playing a systemic role in IGM aetiology. Amongst the 6 significantly different species between paired IGM samples, only the median relative abundance of *C. kroppenstedtii* exceeds 1% in IGM pus (Appendix A). This is not the first time *C. kroppenstedtii* has been associated with IGM: the species has been previously identified from patient sample cultures [34,35,36]. *C. kroppenstedtii* in patient samples is underreported, since routine culture methods are unlikely to detect slow growing *Corynebacterium*, and acid-fast bacillus (AFB) and Periodic acid–Schiff (PAS) stains will not detect *Corynebacterium* either [37]. *Corynebacterium* is also often not described to be tested for in IGM patient samples in the literature [36]. Further molecular studies must be conducted to determine the systemic role of higher *Corynebacterium kroppenstedtii* relative abundance in IGM pus samples.

We observed high interpatient variability within patients with IGM in their different microbial profiles, with MDS only explaining less than 30% of variability with the first two reduced factors (Figure 2b and Appendix A), and large IQR for both alpha-diversity indices (Appendix A). This demonstrates a spread in metagenomic profiles across patients with IGM. This is to be expected, with the demonstrated variability in patient demographics (Table 1). Additionally, as a disease diagnosed by exclusion, with varied clinical presentation, and yet to be determined aetiology, patient heterogeneity is to be expected, in patient demographics and microbial profiles [3,4,5,38]. Further investigations into IGM molecular basis and elucidating the elusive aetiology will be very helpful for classifying the disease into different subtypes. Future studies with larger sample sizes, could consider utilising the metagenomic profiles to categorise meaningful disease subtypes.

An important factor patient variable that interpatient metagenomic variability could be attributed to would be treatment exposure. Previous studies have described steroid treatment affecting microbiomes in different sites, including the lungs, gut and vagina [39,40,41,42]. As a lipid soluble drug, corticosteroids would penetrate fatty breast tissue and could modify pus microbiome in IGM [43]. Antibiotic treatment is widely documented to create great imbalances to the gut microbiome [44,45]. Evidence also describes antibiotic use to affect microbiome at other anatomical sites, and associated with diseases including affected lipid metabolism, inflammation, and auto-immune conditions [46,47,48]. Variability in treatment exposure to antibiotics and/or corticosteroids treatment, and duration from last treatment to sample collection, as observed amongst patients in our study (Table 1), will affect systemic microbiome [47], and could explain some of the interpatient metagenomic variability we observed in our study.

The 19 statistically significant genera observed between paired skin and pus samples from patients with IGM after adjustment for treatment exposure and duration, and correcting for multiple comparisons would be worthy to explore. However, median relative abundance for IGM pus samples does not exceed 5% for any of the genera (Appendix A). Genera with median relative abundance exceeding 1% in IGM pus samples were *Ochrobactrum*, *Delftia*, and *Streptococcus*. *Delftia* genus are Gram-negative rods, rarely involved in human infections, and more often found in environmental sites [49]. All three factors make *Delftia* an unlikely culprit for granuloma formation in breast tissue. However, there is a possibility that *Delftia* genus may contribute to a unique microbiome that makes a patient susceptible to IGM inflammation. *Streptococci*, often implicated as pathogenic in skin infections, is also observed in all paired LM samples at higher relative abundances than that of IGM paired samples (Appendix A) [50,51]. Without significant difference in relative abundance between IGM and LM pus samples, *Streptococci* is not a suitable target for elucidating IGM-specific aetiology or for clinical disease management. *Ochrobactrum* genus is an interesting avenue to explore, as opportunistic pathogens responsible for infectious outbreaks: Ryan and Pembroke (2020) found 128 separate instances, involving 289 unique cases [52]. Furthermore, abscess in non-breast sites (neck, pelvic, pancreatic and retropharyngeal) was a reported symptom of infection [52]. *Ochrobactrum* would also explain poor antibiotic response in IGM given the genus’ resistance to β-lactams (penicillins, cephlasporins and emergingly carbapenem-resistance as well) [52]. However, *Ochrobactrum* is largely an opportunistic pathogen (infected patients are usually immunocompromised with cancer, or diabetes-causing kidney failure, acquiring the infection through catheter or dialysis in clinical settings) [52], whereas the patients with IGM in our study are self-reported without autoimmune conditions, cancer history, or active infectious disease (Autoimmune conditions: Coeliac disease, Type 1 diabetes mellitus, Graves’ disease, Inflammatory bowel disease, Multiple sclerosis, Psoriasis, Rheumatoid arthritis, or Lupus erythematosus; Infectious disease: Tuberculosis [TB], Bacterial Infection, or Fungal Infection) (Table 1). Nonetheless, *Ochrobactrum*, *Delftia*, and *Streptococcus* genera could possibly promote the growth of other organisms responsible for granuloma formation, and are implicated in IGM aetiology. Other significantly different genera and species between paired IGM patient samples have median relative abundances below 1% for both IGM pus and skin samples; as such, they would not serve as meaningful targets for diagnostic, therapeutic, or molecular investigations. 

Patients with IGM self-reported treatment with Cephalexin, Amoxicillin-clavulanate, Clindamycin, Trimethoprim-sulfamethoxazole, and/or Erythromycin largely targets *Streptococcus* and *Staphylococcus* infections, with Erythromycin targeting Gram-positive bacteria that are often implicated in granulomatous inflammations [53,54,55,56,57]. With further information gleaned from metagenomic analysis on the genera and their abundances populating the IGM pus samples, they can serve to guide targeted antibiotic treatment either. The polymicrobial profiles will probably require combination or broad-spectrum antibiotic treatment. However, without definitively identifying a microbial infectious cause, further antibiotic treatment may not be the solution either.

To the best of our knowledge, this is the first study for patients with IGM that provides the duality of intra-patient controls with paired skin and pus samples, on top of the comparison with patients with LM. Furthermore, the use of 16S sequencing is less common for IGM studies, beyond bacteria culture methods, so as to profile both living and dead microbial populations. Metagenomic analysis matching patient samples to previous antibiotic treatment would also identify antibiotic-resistant strains. However, as a rare condition, the number of patients with IGM recruited is limited. The limited LM control patients, as well, means comparisons and conclusions made on the basis of imbalances between IGM and LM patient numbers in our study must be validated in a larger cohort. Future skin sample collections must control for washing and exposure (cloth covering) before sample collection. Subsequent studies should also use swabs moistened with preservation medium or enzymatic lysis buffer, as opposed to dry swabs, for skin sample collection, for improved biomass collection [58]. It must definitely be acknowledged that the microbial load on skin is extremely low, so appropriate sampling and relevant negative controls are extremely crucial. Future work for higher resolution definition of microbial populations in IGM pus and skin samples should absolutely consider 

third generation sequencing platforms to sequence the entire 16S rRNA gene [59]; orshot-gun metagenomics for sequencing the entire microbial genome [60].

## 4. Materials and Methods

### 4.1. Study Population

The study population was adult patients (aged 21 years and above) suspected, diagnosed with or treated for mastitis (IGM or LM) at five participating hospitals in Singapore: National University Hospital (NUH), Tan Tock Seng Hospital (TTSH), KK Women’s and Children’s Hospital (KKH), Singapore General Hospital (SGH), and Changi General Hospital (CGH). Pregnant and breastfeeding women, and lactating and non-lactating women were allowed to participate in this study. Patients who had received or were receiving treatment at the point of recruitment were allowed to participate in the study. Patient information on demographic, lifestyle, reproductive, past and current treatment or environmental exposure variables were collected from questionnaires. All studies were performed in accordance with the Declaration of Helsinki. All participants provided written informed consent. This study was approved by the National Healthcare Group Domain Specific Review Board (reference number: 2017/01057) and the Agency for Science, Technology and Research Institutional Review Board (reference number: 2020-152).

### 4.2. Patient Recruitment and Sample Collection

Adult female patients with IGM or LM were recruited from the five participating hospitals in Singapore between 2018 and 2020. Target recruitment was originally 60 participants per site, totalling 300 participants in the study. During the recruitment time period, 148 patients were screened, 91 patients enrolled in the study, and 89 patients completed the study (2 withdrawals). The main study recruited patients who donated any of the following: blood, saliva, tissue, pus swab, and / or skin swab. In this metagenomic study, a subset of 32 patients provided pus swabs, and / or skin swabs, of which 24 were included for metagenomic analysis (Figure 4 and Appendix A).

Study coordinators were informed by clinicians and physicians of potential mastitis cases, after which, study coordinators would seek consent, collect patient samples and conduct the questionnaire. Patients were subsequently followed up for clinical diagnosis confirmation. Patients who were not subsequently diagnosed as IGM or LM were excluded from the study. Non-lactating patients presenting with mastitis and/or breast abscess were diagnosed as IGM by histological examination of breast core biopsy for non-caseating granulomatous formation. Other conditions were excluded via negative stains for acid-fast bacillus (AFB) to rule out *Mycobacterium tuberculosis* infection, and negative stains for Grocott (methenamine) silver (GMS) stain or Periodic acid–Schiff (PAS) to rule out fungal infections. Histopathology also confirmed non-malignancy. Patients presenting with mastitis and/or breast abscess confirmed to be lactating at the point of recruitment were diagnosed as LM.

Patient samples were collected using DNA/RNA Shield Lysis Tubes w/Swab (Microbe) (catalogue number R1104; Zymo Research, Irvine, CA, USA). Sterile methods were used to collect pus samples from the infected breast, while skin samples were collected from the contralateral (non-infected breast) without site cleansing prior to sample collection. The kit sample collection tubes, that were pre-filled with DNA/RNA Shield and a lysis reagent for microbial samples, were labelled before sample collection. Swab was removed from packaging without touching the applicator tip.

Pus sample collection: Swab was allowed to absorb infected fluid for 30 s while rotating the swab;Skin sample collection: Swab was used to rub skin area for 10 to 15 strokes with moderate pressure. Swab was rotated and sampling was repeated.

Swab was broken along the breakpoint and the applicator head was left in the collection tube. Each collection tube was capped and shaken 10 times to mix the sample collected with pre-filled reagents. 

### 4.3. DNA Extraction and 16S Ribosomal rRNA Sequencing

Total DNA was extracted with ZymoBIOMICS™ 96 MagBead DNA Kit (catalogue number D4306; Zymo Research, Irvine, CA, USA). Samples were mechanically lysed with beads beating at maximum speed for 20 minutes. DNA was purified from lysed samples with magnetic beads (ZymoBIOMICS™ MagBinding Beads) and eluted with ZymoBIOMICS™ DNase/RNase Free Water. The 16S ribosomal RNA (rRNA) gene amplification and sequencing has been previously described [61]. Briefly, the V2-V6 region (723 bp PCR product) of the 16S rRNA gene was amplified via PCR with the HotStar HiFidelity Polymerase Kit (catalogue number 202602; QIAGEN, Hilden, Germany). Primers used and their sequences are shown in Table 3. Appendix A shows the 16S rRNA gene map and the loci of the primers used along the gene. PCR parameters were initial denaturation at 95 °C for 5 minutes, followed by 35 PCR cycles of

Denaturation at 95 °C for 30 seconds;Annealing at 59 °C for 30 seconds; andExtension at 72 °C for 1 minute.

The final elongation was at 72 °C for 6 minutes. QIAquick PCR Purification Kit (catalogue number 28106; QIAGEN, Hilden, Germany) was used to purify 700–1000 bases PCR products. Amplicons were sheared with Covaris Adaptive Focused Acoustics™ to approximately 180 base fragments. The 16S libraries were prepared using NEBNext^®^ Ultra™ II DNA Library Prep Modules for Illumina^®^ (catalogue number E7645L; New England Biolabs, Ipswich, MA, USA) according to the manufacturer’s protocol. Multiplexing Sample Preparation Oligonucleotide Kit from Illumina, Inc. (San Diego, CA, USA) was used to label the DNA sequencing libraries with different multiplex indexing barcodes [61]. The 200 pM concentration 16S libraries were multiplexed paired-end sequenced on HiSeq4000 with 2 × 150 bp paired-end reads.

### 4.4. Sequencing Data Processing

Processing of sequencing data has been previously described [61]. Briefly, data was de-multiplexed and sequencing reads that passed Illumina’s purity/chastity filter (PF = 0) were converted to FASTQ format. Trailing bases with quality scores lower or equal to 2 were subsequently filtered and trimmed from the 3′ end; read pairs containing reads shorter than 120 bases were removed [62]. An adapted expectation maximization iterative reconstruction of genes from the environment (EMIRGE) assembly, was used to reconstruct the amplicons of the shorter-read dataset (EMIRGE was originally designed for whole genome datasets) by only performing analysis with top 100,000 average quality reads [61]. Reconstructed sequences were trimmed according to the region amplified by the selected primers; reconstructed sequences were searched against the complete Greengenes database (dated 1 May 2019; gg_13_5/gg_13_5_otus.tar.gz/rep_set/99_otus.fasta, from https://greengenes.secondgenome.com/?prefix=downloads/greengenes_database/gg_13_5/, accessed on 19 November 2020) using Basic Local Alignment Search Tool (BLAST) [63]. The top hit on BLAST by lowest E-value, highest bit-score, highest percent identity and longest alignment length, in that particular order, was used to classify the sequence [61]. Percentage identity for genus classification must be ≥95% (≥97% for species classification) to be classified.

### 4.5. Statistical Analysis

Statistical analysis was performed using a modified microbiome abundance analysis-suite (https://github.com/BetaCollins/MBAA-Suite, accessed on 16 December 2020) customized R resource that contains the analysis functions of 16S pipeline (https://github.com/CSB5/GERMS_16S_pipeline, accessed on 16 December 2020) developed at the Genome Institute of Singapore, designed for Illumina shotgun sequencing of 16S rRNA amplicon sequences [61]. Microbial populations were evaluated at genus and species taxonomy levels.

Microbial populations of pus samples were compared between IGM and LM controls and repeated in skin samples. Shannon (Equation (1)) and Simpson (Equation (2)) indices were used to measure alpha-diversity.
(1)$$Shannon\;Index\;\left(H\right)=-\overset{s}{\underset{i=1}{\sum }}p_{i}\ln p_{i}$$
(2)$$Simpson\;Index\;\left(D\right)=\frac{1}{\sum_{i=1}^{s}p_{i}{}^{2}}$$

Multidimensional scaling was used for visualisation of Bray−Curtis distance between the microbial profiles of the IGM and LM sample types. Permutational multivariate analysis of variance (PERMANOVA) was used to make comparisons of mastitis and sample types centroids and dispersion within groups; statistical significance was defined by measure of space not equivalent for all groups. Relative abundance of *Corynebacterium* genera was evaluated in pus and skin samples from patients with IGM and LM. Paired sample analysis with Wilcoxon signed-rank test, compared *Corynebacterium* genera relative abundance in pus and skin samples from the same patient in patients with IGM. Microbiome multivariable association was performed with linear models using MaAsLin 2 (https://github.com/biobakery/biobakery/wiki/maaslin2, accessed on 25 March 2022) [64] to compare the microbial populations between IGM and LM patient samples; paired sample analyses, and adjusted analyses with antibiotic treatment, and duration after antibiotic treatment as confounders were also performed.

All analyses were performed with R (v4.0.2) unless otherwise stated. Datasets utilised in analysis can be made available from the corresponding author upon reasonable request, within limitations of the study Institutional Review Board (IRB).

## 5. Conclusions

Microbial profiles are unique between patients with IGM and LM. Our study has demonstrated statistically significant difference in the microbial communities of IGM pus, IGM skin, LM pus, and LM skin samples. While *Corynebacterium* genus was not significantly different between paired pus and skin samples, the *Corynebacterium kroppenstedtii* species was, and so were 19 genera and 3 other species. However, the inter-patient variability and polymicrobial microbiome of IGM pus samples cannot implicate specific genus or species of bacteria in an infectious cause for IGM. Subsequent investigations with improved sampling methods, and employment of advanced metagenomics methods can achieve higher resolution for categorical definition of microbial populations in IGM pus and skin samples. Studies with larger patient numbers, backed by molecular experiments, are needed to confirm or discard an infectious aetiology for IGM.

## Figures and Tables

**Figure 1 ijms-24-01042-f001:**
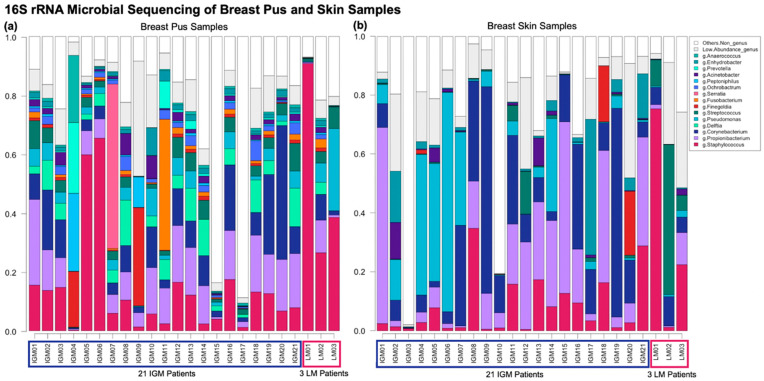
Bar graph of genus taxonomic profiles of breast (**a**) pus and (**b**) skin samples. A metagenomic analysis compared the relative abundances of bacterial taxa of pus samples from 21 patients with idiopathic granulomatous mastitis (IGM) and three control patients with lactating mastitis (LM) (LM01, LM02, and LM03). Each bar represents a (**a**) pus or (**b**) skin sample from a patient. The genus level taxonomic units with relative abundance ≥ 1% are represented, with corresponding colours indicated in the shared figure key for (**a**,**b**) on the right. Genus taxonomic units with relative abundance < 1% are collapsed into “Low Abundance”, and taxonomic units higher than genus level classification are collapsed into “Others”. (**a**) Genera *Staphylococcus* dominates the three LM pus samples (LM1-3): 90.8%, 26.5%, and 38.5%, respectively. Bacteria profiles in IGM pus samples differ from LM, without a dominant genus across pus samples, and greater variability between patients; (**b**) No dominant genus is observed across either the IGM or LM skin samples, collectively or as separate groups. *Corynebacterium* contributes more to skin than pus microbiota in both IGM and LM samples, although *Corynebacterium* appears at higher relative abundances for IGM skin samples than LM skin samples. Again, inter-patient variability is considerable. IGM: Idiopathic granulomatous mastitis; LM: Lactational mastitis.

**Figure 2 ijms-24-01042-f002:**
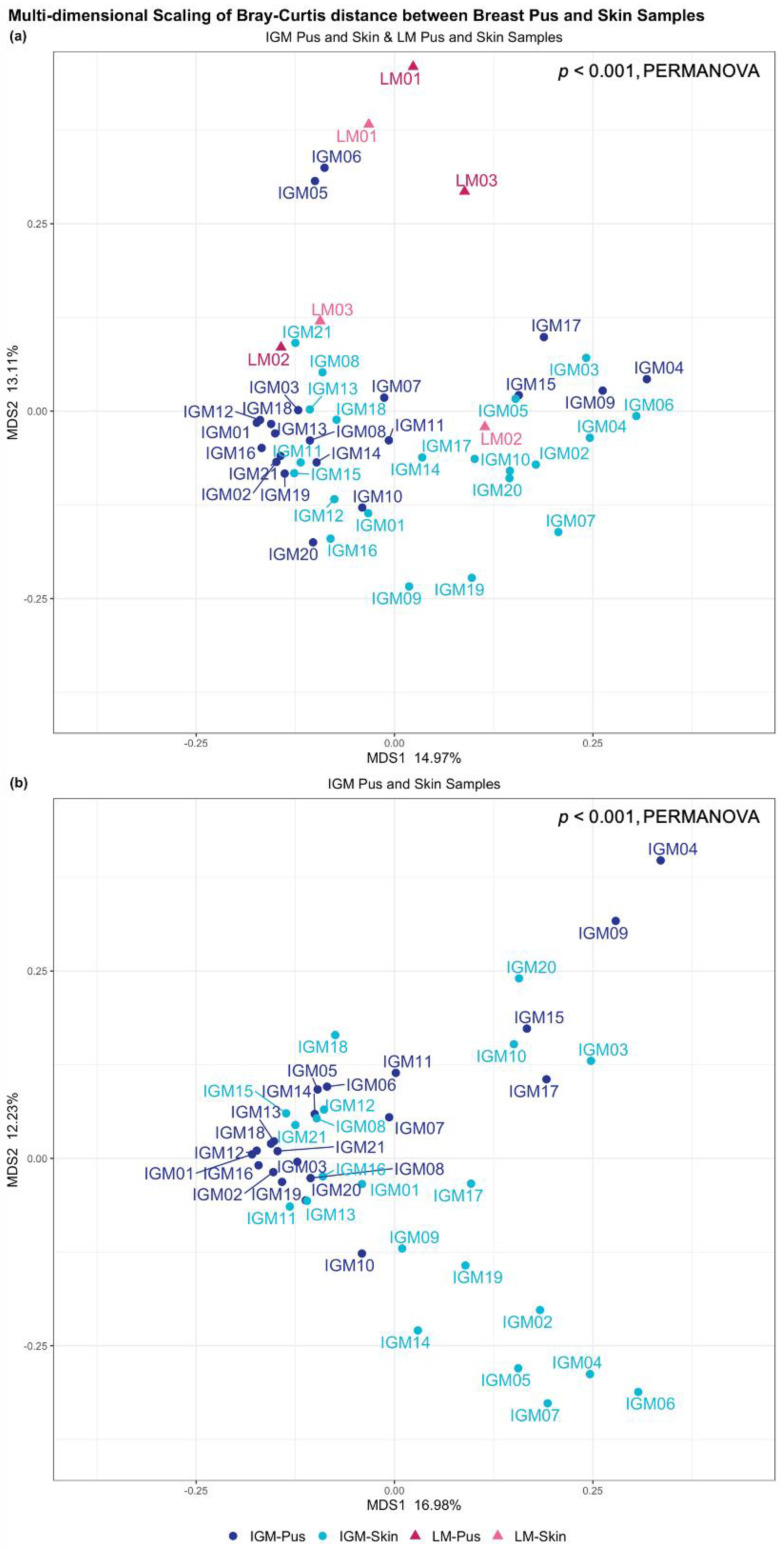
Multi-dimensional scaling (MDS) (principal coordinate analysis (PCoA)) of the Bray−Curtis distance between breast pus and skin samples from (**a**) patients with idiopathic granulomatous mastitis (IGM) and control patients with lactating mastitis (LM), and (**b**) patients with IGM alone. Each point represents a pus or skin sample from a patient. The 21 patients with IGM are represented as dots and the 3 control patients with LM are represented as triangles, with corresponding colours indicating sample type shown in the shared figure key for (**a**,**b**) at the bottom. (**a**) First two MDS of Bray−Curtis distance between breast pus and skin samples for patients with IGM and LM only explains 14.97% and 13.11% of the variability, respectively, and collectively explains 28.08% of the samples’ variability. Despite no obvious clustering, the different patient and sample types affected genera diversity significantly (*p* < 0.001, PERMANOVA); (**b**) First two MDS of Bray−Curtis distance between breast pus and skin samples for patients with IGM only explains 16.98% and 12.23% of the variability, respectively, and collectively explains 29.21% of the samples’ variability. Despite no obvious clustering, the different patient and sample types affected genera diversity significantly (*p* < 0.001, PERMANOVA). IGM: Idiopathic granulomatous mastitis; LM: Lactational mastitis; MDS: Multi-dimensional scaling; *p*: *p*-value; PERMANOVA: Permutational multivariate analysis of variance.

**Figure 3 ijms-24-01042-f003:**
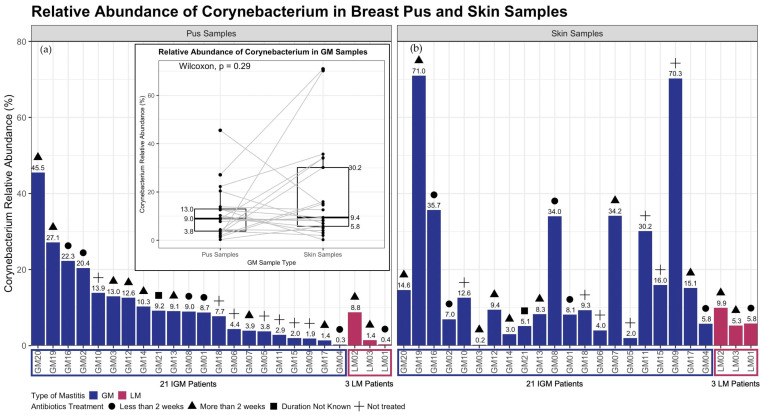
Relative abundance of *Corynebacterium* genus in breast (**a**) pus and (**b**) skin samples from patients with idiopathic granulomatous mastitis (IGM) and control patients with lactating mastitis (LM). The left-end bars in both panels represent the 21 patients with IGM, and the right-end bars in both panels represent the 3 control patients with LM, with corresponding colours indicating mastitis type shown in the bottom-left shared figure key for (**a**,**b**). The percentage of *Corynebacterium* genus relative abundance is indicated above each bar, presented to 1 decimal place. Duration of antibiotic treatment prior to sample collection are presented as symbols above each bar, as four categories: Less than 2 weeks before sample collection, more than 2 weeks before sample collection, missing duration, and no antibiotic treatment. The corresponding symbols are also indicated in the bottom-left shared figure key for (**a**,**b**). (**a**) For the left panel of pus samples, the patients are arranged in decreasing relative abundance of *Corynebacterium* genus in pus samples within mastitis type; (**b**) For the right panel of skin samples, the patients are arranged following patient order in (**a**), i.e., decreasing relative abundance of *Corynebacterium* genus in pus samples within mastitis type. Distribution of relative abundance of *Corynebacterium* genus in breast pus and skin samples from 21 patients with IGM is also displayed in (**a**). Median *Corynebacterium* relative abundance in IGM pus samples is 9% (interquartile range = 3.8–13.0%), compared to 9.4% (interquartile range = 5.8–30.2%) in IGM skin samples. Paired Wilcoxon sign ranked test found no significant difference in *Corynebacterium* relative abundance between paired IGM skin and pus samples (*p* = 0.29). IGM: Idiopathic granulomatous mastitis; LM: Lactational mastitis; *p*: *p*-value; Wilcoxon: Wilcoxon paired sign ranked test.

**Figure 4 ijms-24-01042-f004:**
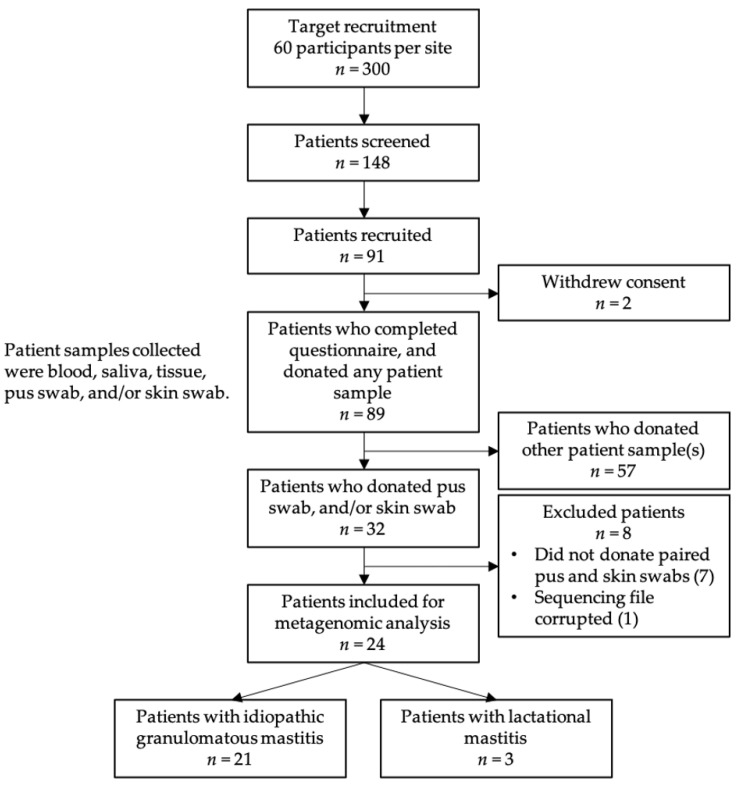
Patient recruitment, inclusion and exclusion flowchart.

**Table 1 ijms-24-01042-t001:** Description of analytical cohort of idiopathic granulomatous mastitis (IGM) and lactational mastitis (LM) patients.

	Total *n* = 24	IGM *n* = 21	LM *n* = 3	*p*-Value ^1^
**Demographics**				
Median age at diagnosis (years, IQR)	34 (30.50–40.25)	34 (31.00–41.00)	31 (28.00–34.00)	0.358

Recruitment site (n, %)				0.422
KKH	13 (54.17)	10 (47.62)	3 (100.00)	
SGH	9 (37.50)	9 (42.86)	0 (0)	
NUH	2 (8.33)	2 (9.52)	0 (0)	

Year of diagnosis (n, %)				0.546
2014–2017	6 (25.00)	6 (28.57)	0 (0)	
2018–2019	18 (75.00)	15 (71.43)	3 (100.00)	

Ethnicity (n, %)				0.308
Chinese	15 (62.50)	14 (66.67)	1 (33.33)	
Malay	4 (16.67)	3 (14.29)	1 (33.33)	
Indian	2 (8.33)	2 (9.52)	0 (0)	
Other	3 (12.50)	2 (9.52)	1 (33.33)	

Body mass index (kg/m^2^, IQR)	26.29 (22.34–30.83)	26.29 (24.61–31.22)	22.19 (22.03–23.91)	0.106

Education level (n, %)				0.727
Primary	2 (8.33)	2 (9.52)	0 (0)	
Secondary	5 (20.83)	4 (19.05)	1 (33.33)	
Pre-University	8 (33.33)	6 (28.57)	2 (66.67)	
Undergraduate	6 (25.00)	6 (28.57)	0 (0)	
Graduate	3 (12.50)	3 (14.29)	0 (0)	

**Patient characteristics**				
Pregnant, at time of sample collection (n, %)				
No	24 (100.00)	21 (100.00)	3 (100.00)	

Lactating, at time of sample collection (n, %)				<0.001
Yes	3 (12.50)	0 (0)	3 (100.00)	
No	21 (87.50)	21 (100.00)	0 (0)	

Previously Pregnant (n, %)				1
Yes	21 (87.50)	18 (85.71)	3 (100.00)	
No	3 (12.50)	3 (14.29)	0 (0)	

Number of children (n, %)				0.769
0	3 (12.50)	3 (14.29)	0 (0)	
1	13 (54.17)	10 (47.62)	3 (100.00)	
2	6 (25.00)	6 (28.57)	0 (0)	
3	1 (4.17)	1 (4.76)	0 (0)	
4	1 (4.17)	1 (4.76)	0 (0)	

Time since last childbirth (n, %)				0.091
No children	3 (12.50)	3 (14.29)	0 (0)	
Less than 2 years	8 (33.33)	5 (23.81)	3 (100.00)	
Between 3 and 5 years	10 (41.67)	10 (47.62)	0 (0)	
More than 5 years	3 (12.50)	3 (14.29)	0 (0)	

Smoking (n, %)				1
Yes	5 (20.83)	5 (23.81)	0 (0)	
No	19 (79.17)	16 (76.19)	3 (100.00)	

Chronic Illness ^2^ Diagnosis (n, %)				1
Yes	4 (16.67)	4 (19.05)	0 (0)	
No	20 (83.33)	17 (80.95)	3 (100.00)	

Autoimmune Conditions ^3^ (n, %)				
No	24 (100.00)	21 (100.00)	3 (100.00)	

Previous Infectious Disease ^4^ Diagnosis (n, %)				
No	24 (100.00)	21 (100.00)	3 (100.00)	

Previous Cancer Diagnosis (n, %)				
No	24 (100.00)	21 (100.00)	3 (100.00)	

**Treatment for mastitis**				
Any treatment (n, %)				1
Yes	21 (87.50)	18 (85.71)	3 (100.00)	
No	3 (12.50)	3 (14.29)	0 (0)	

Type of treatment (n, %)				0.185
Antibiotic treatment ^5^ only	9 (42.86)	6 (33.33)	3 (100.00)	
Antibiotic and steroid treatment ^6^ only	8 (38.10)	8 (44.44)	0 (0)	
Other type of treatment	4 (19.05)	4 (22.22)	0 (0)	

Duration between antibiotic treatment and sample collection (n, %)				0.782
Less than 2 weeks	7 (29.17)	6 (28.57)	1 (33.33)	
More than 2 weeks	9 (37.50)	7 (33.33)	2 (66.67)	
Duration missing	1 (4.17)	1 (4.76)	0 (0)	
Did not receive antibiotic treatment	7 (29.17)	7 (33.33)	0 (0)	

^1^ Comparison between patients with IGM and LM using Kruskal–Wallis test for continuous variables, and Fisher’s exact test for categorical variables. ^2^ Chronic illnesses: Heart attack, stroke, or high blood pressure. ^3^ Autoimmune conditions: Coeliac disease, type 1 diabetes mellitus, Graves’ disease, inflammatory bowel disease, multiple sclerosis, psoriasis, rheumatoid arthritis, or lupus erythematosus. ^4^ Infectious diseases: Tuberculosis, bacterial infection, or fungal infection. ^5^ Known antibiotics treatment: Cephalexin, Amoxicillin-clavulanate, Clindamycin, Trimethoprim-sulfamethoxazole, and/or Erythromycin. ^6^ Known steroid treatment: Corticosteroids, and/or methotrexate. IGM: Idiopathic granulomatous mastitis; LM: Lactational mastitis; IQR: Interquartile range; KKH: KK Women’s and Children’s Hospital; NUH: National University Hospital; SGH: Singapore General Hospital; NCCS: National Cancer Centre Singapore; TTSH: Tan Tock Seng Hospital.

**Table 2 ijms-24-01042-t002:** Statistically significant (after adjustments and correcting for multiple comparisons) genera and species identified from general linear models for determining multivariable association between sample type, covariates and microbial metagenomic features in paired pus and skin samples from patient with idiopathic granulomatous mastitis (IGM).

	*n* (*n*, not 0)	Crude				Adjusted ^1^			
		β ^2^	SD ^2^	*p*-Value ^3^	q-Value ^4^	β ^2^	SD ^2^	*p*-Value ^3^	q-Value ^4^
**Genus**									
*Ochrobactrum*	42 (40)	−2.770	0.154	<0.001	<0.001	−2.770	0.156	<0.001	<0.001 ***
*Delftia*	42 (42)	−2.469	0.196	<0.001	<0.001	−2.469	0.196	<0.001	<0.001 ***
*Anaerobacillus*	42 (28)	−2.470	0.247	<0.001	<0.001	−2.470	0.247	<0.001	<0.001 ***
*Gordonia*	42 (42)	−1.431	0.159	<0.001	<0.001	−1.431	0.159	<0.001	<0.001 ***
*Methylobacterium*	42 (16)	1.361	0.206	<0.001	<0.001	1.361	0.206	<0.001	<0.001 ***
*Fusobacterium*	42 (39)	−1.577	0.249	<0.001	<0.001	−1.577	0.249	<0.001	<0.001 ***
*Sphingobium*	42 (15)	1.111	0.232	<0.001	0.003	1.111	0.225	<0.001	0.003 **
*Alkanindiges*	42 (32)	−1.436	0.313	<0.001	0.006	−1.436	0.297	<0.001	0.004 **
*Streptococcus*	42 (42)	−0.884	0.168	<0.001	0.002	−0.884	0.168	<0.001	0.005 **
*Achromobacter*	42 (16)	0.832	0.167	<0.001	0.003	0.832	0.167	<0.001	0.009 **
*Capnocytophaga*	42 (15)	1.131	0.249	<0.001	0.006	1.131	0.249	<0.001	0.022*
*Mycobacterium*	42 (22)	1.404	0.350	<0.001	0.007	1.404	0.344	<0.001	0.022 *
*Novosphingobium*	42 (13)	0.627	0.142	<0.001	0.006	0.627	0.142	<0.001	0.024 *
*Peptoniphilus*	42 (42)	−1.174	0.271	<0.001	0.008	−1.174	0.271	<0.001	0.025 *
*Rothia*	42 (37)	−1.038	0.239	<0.001	0.008	−1.038	0.239	<0.001	0.025 *
*Finegoldia*	42 (42)	−1.152	0.271	<0.001	0.008	−1.152	0.271	<0.001	0.027 *
*Burkholderia*	42 (15)	0.699	0.178	<0.001	0.008	0.699	0.181	<0.001	0.028 *
*Roseomonas*	42 (25)	1.328	0.340	<0.001	0.008	1.328	0.348	<0.001	0.031 *
*Anaerococcus*	42 (42)	−0.808	0.221	0.002	0.027	−0.808	0.216	<0.001	0.036 *
*Agrobacterium*	42 (22)	1.336	0.378	0.001	0.019	1.336	0.380	0.001	0.065
*Hydrogenophaga*	42 (11)	0.914	0.276	0.002	0.030	0.914	0.263	0.001	0.069
*Peptostreptococcus*	42 (9)	0.768	0.226	0.002	0.035	0.768	0.226	0.003	0.120
*Kocuria*	42 (18)	0.877	0.280	0.003	0.043	0.877	0.284	0.004	0.144
*Dermabacter*	42 (17)	0.996	0.318	0.003	0.043	0.996	0.324	0.004	0.145
**Species**									
*Acinetobacter schindleri*	42 (21)	−1.571	0.226	<0.001	<0.001	−1.571	0.226	<0.001	<0.001 ***
*Rothia mucilaginosa*	42 (30)	−1.472	0.285	<0.001	<0.001	−1.472	0.289	<0.001	0.002 **
*Lactobacillus iners*	42 (13)	0.695	0.183	<0.001	0.021	0.695	0.183	<0.001	0.039 *
*Corynebacterium kroppenstedtii*	42 (38)	−0.971	0.245	<0.001	0.019	−0.971	0.245	<0.001	0.053
*Roseomonas mucosa*	42 (24)	1.071	0.321	0.002	0.033	1.071	0.329	0.002	0.116
*Kocuria rhizophila*	42 (12)	0.780	0.246	0.003	0.045	0.780	0.251	0.004	0.162

^1^ Adjusted for antibiotic treatment (received and did not receive antibiotics treatment), and duration after antibiotics treatment (samples collected less than 2 weeks after treatment vs. more than 2 weeks after treatment). ^2^ β: General linear model beta-estimates; SD: Standard deviation. Beta-estimates and standard deviations presented are for pus samples as the reference group against which paired skin samples were compared. ^3^ Comparison between paired pus and skin samples from patients with IGM using general linear models for determining multivariable association between sample type, covariates and microbial metagenomic features. ^4^ q-value represents the adjusted *p*-value for multiple comparisons using Benjamini–Hochberg correction (1995). ***: q-value < 0.001; **: 0.001 ≤ q-value < 0.01; *: 0.01 ≤ q-value < 0.05.

**Table 3 ijms-24-01042-t003:** Primer set used for 16S rRNA gene amplification.

Primer	Sequences (5′-3′)
338F	ACTYCTACGGRAGGCWGC
1061R	CRRCACGAGCTGACGAC

## Data Availability

The datasets used and/or analysed during the current study are available from the corresponding author on reasonable request, within limitations of the study Institutional Review Board (IRB).

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
