# Peer review of "Profiling Microbial Communities in Idiopathic Granulomatous Mastitis"

_ijms, 2023, doi:10.3390/ijms24021042_

Round 1

Reviewer 1 Report

16S rRNA sequencing for profiling microbial communities in idiopathic granulomatous mastitis

This papers describes an important issue. Before publishing the manuscript, the following comments should be considered.

L 65 Hypothesised aetiologies of IGM have include a possible microbial cause. Please include the microbial cause.

L 88-96 The research question is faintly understandable. I request to add previously what have done and hat have not been done and write the aims and objectives of the present study clearly.

L 154-155 write the primer sequence in a table. Methodology should be write more clearly.

Figure 2 is not legible. Please revise the figure.

Discussion should be revised. Emphasis should be given on key findings.

The conclusions are very brief. Please mention key findings in this section and some recommendations for further work.

Reviewer 2 Report

See my comments in the attached pdf. Use adobe reader to see my revision

Round 2

Reviewer 2 Report

Thank you for your corrections. All my comments were successfully solved.